# Group vs. Individual Treatment for Acute Insomnia: A Pilot Study Evaluating a “One-Shot” Treatment Strategy

**DOI:** 10.3390/brainsci7010001

**Published:** 2016-12-23

**Authors:** Pam Boullin, Christina Ellwood, Jason G. Ellis

**Affiliations:** Northumbria Sleep Research Laboratory, Northumbria University, Newcastle-upon-Tyne NE18ST, UK; Pamela.Boullin@nuth.nhs.uk (P.B.); christina.ellwood@northumbria.ac.uk (C.E.)

**Keywords:** acute insomnia, CBT-I, treatment, group therapy

## Abstract

Background: Despite undeniable evidence for the efficacy and effectiveness of Cognitive Behaviour Therapy for Insomnia (CBT-I), the potential for its widespread dissemination and implementation has yet to be realised. A suggested reason for this is that traditional CBT-I is considered too burdensome for deployment, in its current form, within the context of where it would be most beneficial—Primary Care. One strategy, aimed to address this, has been to develop briefer versions of CBT-I, whilst another has been to deliver CBT-I in a group format. An alternative has been to attempt to address insomnia during its acute phase with a view to circumventing its progression to chronic insomnia. The aim of the present study was to compare a brief version of CBT-I (one-shot) when delivered individually or in groups to those with acute insomnia. Method: Twenty-eight individuals with acute insomnia (i.e., meeting full DSM-5 criteria for insomnia disorder for less than three months) self-assigned to either a group or individual treatment arm. Treatment consisted of a single one-hour session accompanied by a self-help pamphlet. Subjects completed measures of insomnia severity, anxiety and depression pre-treatment and at one-month post-treatment. Additionally, daily sleep diaries were compared between pre-treatment and at the one-month follow up. Results: There were no significant between group differences in treatment outcome on any sleep or mood measures although those in the group treatment arm were less adherent than those who received individual treatment. Furthermore, the combined (group and individual treatment arms) pre-post test effect size on insomnia symptoms, using the Insomnia Severity Index, was large (*d* = 2.27). Discussion: It appears that group treatment is as efficacious as individual treatment within the context of a “one shot” intervention for individuals with acute insomnia. The results are discussed with a view to integrating one-shot CBT-I in Primary Care.

## 1. Introduction

Insomnia is a significant public health concern affecting approximately 30%–36% of the population in its subsyndromal form and between 8% and 15% in its syndromal, or chronic, form [1,2]. Not only are the costs, both direct and indirect, associated with insomnia considerable but recent research points to insomnia being a risk factor for the development and/or worsening of a myriad of other illnesses and conditions [3,4,5]. Moreover, chronic insomnia appears to be persistent with studies demonstrating both low natural remission rates [6] and high recurrence rates [7] over time. As such, there is a definite need to address insomnia at both individual and societal levels.

Cognitive Behavioural Therapy for Insomnia (CBT-I), traditionally delivered individually face-to-face over the course of 6–8 weeks, is now considered the first-line treatment option for individuals with chronic insomnia [8,9]. Through a series of taught techniques, the aim of CBT-I is to increase the drive to sleep whilst eliminating any conditioned arousal to the bedroom and bedtime routine, reducing sleep-related catastrophic worry and concerns and identifying and challenging any pre-existing sleep-related dysfunctional attitudes and beliefs. In terms of its potency, there are now a large number of meta-analyses (see Figure 1) which not only demonstrate the efficacy and effectiveness of CBT-I, in the main, but also show that it can be successfully delivered in: (a) a group context; (b) via digital technology; and (c) to individuals with a range of physical and psychological co-morbidities [10]. That said, there are several issues that hamper the widespread dissemination and implementation of CBT-I. Notably, and despite its relatively short duration compared to other psychotherapeutic techniques, traditional CBT-I is often considered too burdensome in its current format by patients [11,12] and this is reflected in the relatively high levels of attrition and non-adherence, especially in clinical settings [13]. To that end, there have been several attempts to modify traditional CBT-I into abbreviated versions [14,15,16,17] to address these issues of patient burden. Whilst these studies have demonstrated good efficacy and effectiveness, the widespread implementation and dissemination of CBT-I has yet to be realized [18].

An alternative perspective, highlighted by Ellis and colleagues in 2012, is to circumvent the development of chronic insomnia by attempting to treat it during its acute phase (i.e., within the first three months of manifesting) [38]. The rationale for this being that treatment during the acute phase should be even less burdensome for the patient and faster to administer due to: (i) less conditioned arousal to the bedroom and pre-sleep routine; and (ii) a less realised self-schemata of having “insomnia” which would be evidenced by lower levels of sleep-related catastrophic worry and sleep-related dysfunctional thinking. Moreover, as the intimate link between insomnia and depression appears to become realised during the acute phase of insomnia, Ellis and colleagues argue a need to treat acute insomnia in its own right in an effort to aid in the prevention of depression [39]. To that end, Ellis, Cushing and Germain [40] created a brief “one-shot” intervention for individuals with acute insomnia. The one-shot consisted of a self-help pamphlet and a single 60–70 min face-to-face treatment session. The pamphlet aimed to identify and address sleep-related dysfunctional thinking, through education about sleep, provide techniques to distract from intrusive worrisome thoughts at night and provide the guidelines for sleep-related stimulus control (using the bedroom for sleep and sex alone and leaving the bedroom if unable to sleep). During the single treatment session a personalised sleep “restriction” protocol is created which focuses on maximising sleep efficiency (the amount of time in bed spent asleep) and increasing the depth and quality of sleep. They tested the efficacy and effectiveness of the intervention in a Randomised Controlled Trial with 40 individuals with acute insomnia. One month after delivery, 60% of those in the treatment arm had remitted whereas only 15% of those in the control group had remitted. Further, the authors showed moderate effect sizes for reducing the time it took to get off to sleep (*d* = 0.77), the amount of time awake during the night (*d* = 0.71) and a relative increase in sleep efficiency (*d* = 0.69). Interestingly, three months following treatment, 73.3% of those in the treatment arm had naturally remitted. Despite the success of this brief intervention there still remains a need to determine whether it can be delivered even more efficiently, without losing its efficacy, when considering the increasing financial and practical demands on healthcare services [41].

The aim of the present study was to examine the impact of group therapy versus individual therapy for the single shot treatment in individuals with acute insomnia. In line with the meta-analytic data from group delivered traditional CBT-I [27], it was expected that the one-shot within the context of group therapy would be as impactful, both in terms of efficacy and effectiveness, than when delivered on an individual basis.

## 2. Method

### 2.1. Participants

Subjects were recruited using two advertising posters, which were displayed for approximately 90 days each in community areas (e.g., libraries, and community centres) in two demographically similar regions in the northeast United Kingdom. The posters were similar in design, asking for individuals who had been suffering from acute insomnia to contact the research team with regard to a new treatment study, except one poster stated the treatment session would be in groups whilst the other stated that the treatment would be delivered individually. Each poster had a different contact email address so the research team could track levels of interest for each treatment arm. In order to be eligible participants had to be: (i) between the ages of 18 and 60 years old; (ii) within the first three months of having insomnia; (iii) no history or current experience of CBT-I; and (iv) not currently using sleep medication. Individuals who reported a suspected, untreated or unstable chronic illness or a diagnosis, whether treated and stable, or not, of migraine, Post Traumatic Stress Disorder, epilepsy, seizures, or Parasomnia were excluded from taking part. With respect to the diagnosis of acute insomnia, in line with DSM-5 criteria for Insomnia Disorder, subjects had to report dissatisfaction with their sleep characterised as either a difficulty getting off to sleep, staying asleep or waking earlier than required. Further, the problem should exist despite adequate opportunity to sleep and should occur for, at a minimum, three nights per week. Finally, subjects had to report that the insomnia results in significant impairment to daytime functioning or mood. Within the context of acute insomnia, subjects had to report that the problem had been present for between two weeks and three months.

### 2.2. Procedure

Potential subjects that contacted the research team, from the email addresses on the poster, were provided an information sheet, which outlined the commitment required by the subject (i.e., two visits to the sleep research laboratory and the completion of questionnaires every week and sleep diaries every day) in addition to the inclusion and exclusion criteria. Those who were eligible and still wanted to take part were screened to ensure they met inclusion criteria and if still eligible were asked to provide informed consent before being enrolled on to the respective arm of the study. Enrolled subjects were sent a weeklong sleep diary and the questionnaires (Insomnia Severity Index [42], Patient Health Questionnaire—9 [43], and General Anxiety Disorder—7 [44]), for baseline assessment, and were provided an appointment date and time, at the sleep research laboratory, to attend the treatment session approximately a week following enrolment. Subjects were instructed to bring the completed questionnaires and sleep diary with them to the session. The procedure for the “one-shot” insomnia treatment is published elsewhere [40]. However, the group therapy subjects were treated in groups of 4. Two Master’s level psychologists delivered treatment and both were trained and supervised by an experienced CBT-I therapist (JGE). During the treatment session subjects’ baseline sleep diaries were used to create personalised sleep plans to follow for the duration of the study. From the personalised sleep plan the average amount of sleep obtained per night (mean Total Sleep Time (TST)) was derived from the baseline sleep diary. TST was then set at the amount of time the individual was required to be in bed each night over the following week (Prescribed Time In Bed (PTIB)) as long as the average was a minimum five hours in duration. If the average from the sleep diaries was less than five hours then PTIB was set at five hours. The subjects, following instruction, and then checked by the investigators during the treatment session, calculated their PTIB. From this point subjects “anchored” their Prescribed Time Out of Bed (PTOB), usually to the time they had to be up in the morning owing to work or social commitments, and then subjects worked backwards to derive their Prescribed Time To Bed (PTTB) for the week. Subjects were told to stick with their personalised sleep plan even on non-work days. At the session subjects were also told that following a week of this new plan they were to titrate their prescription based upon their Sleep Efficiency averages from the week using the following rules: less than 85% SE they were to increase their PTTB by 15 min (i.e., from 12:45 a.m. to 1:00 a.m.), a SE between 85% and 90% to stay with their current schedule and a SE of more than 90% they were to reduce their PTTB by 15 min (i.e., from 12:45 a.m. to 12:30 a.m.). Again, subjects were told not to go below the minimum threshold of five hours PTIB. Finally, subjects were told to titrate at the end of each week for the remaining duration of the study. On completion of the treatment session subjects were provided four weeklong sleep diaries to complete over the duration of the study and a follow-up appointment was made for them to return the diaries and complete the same questionnaires that they completed at baseline. On completion of the study, subjects were thanked, debriefed and offered additional support (i.e., a full six week course of CBT-I) if necessary. The study had been granted ethical approval by Northumbria University Ethics Committee (SUB33_300315 & SUB17_180315). 

### 2.3. Measures

The Insomnia Severity Index (ISI) is a 7-item self-report indicator of the presence and severity of insomnia. It takes approximately 5 min to complete. The scale has been used extensively and has excellent psychometric properties [42]. Furthermore, the ISI has been shown to be sensitive to change following intervention [45]. For the purposes of the present study, subjects were asked to complete the ISI on the basis of the last week. Scores range between 0 and 28 with higher scores indicating higher insomnia symptomology.

The Patient Health Questionnaire (PHQ 9) is a 9-item self-report questionnaire that examines depression severity. It takes approximately 5 min to complete. The scale is suitable for assessing depression and monitoring change following intervention and has been used extensively in health-related research. It has excellent psychometric properties and is sensitive to change following intervention. For the purpose of the present study the reporting period was matched to that of the other questionnaires and subjects reported on the basis of the last week. For the purposes of analysis, item 3 “Trouble falling or staying asleep, or sleeping too much” was excluded. As such scores ranged from 0 to 24 with higher scores indicating higher depression symptomology.

The GAD-7 is a 7-item self-report questionnaire that examines anxiety severity. It takes approximately 3 min to complete. The scale is suitable for assessing Generalized Anxiety Disorder and monitoring change following intervention and has been used extensively in primary care research. Scores on the GAD-7 range between 0 and 21 with higher scores indicating higher anxiety symptomology. It has excellent psychometric properties and is sensitive to change following intervention. As with the ISI and PHQ-9, subjects were asked to report on the basis of the previous week.

The Consensus Sleep Diary [46] contains 10 items relating to the previous night’s sleep. Subjects were asked to complete the sleep diary each morning over the duration of the study, approximately 20–40 min after awakening. From the sleep diary the variables of interest related to the main symptoms of insomnia and were derived from the data averaged over the previous week—Sleep Latency (the time elapsed between intending to sleep and sleeping), Wake After Sleep Onset (the duration of time awake over the course of the night not including sleep latency), Sleep Efficiency (the amount of time spent in bed asleep over the duration of time spend in bed overall—expressed as a percentage) and Total Sleep Time (the amount of sleep obtained). 

### 2.4. Analytic Strategy

Baseline differences in demographic characteristics and self-reported sleep were examined using a series of *t*-tests. Following checks for normality and homogeneity, a mixed ANOVA was used to examine between and within group differences on the primary outcome measure (i.e., ISI scores). Further, pre-post treatment change scores were used in a multivariate ANOVA to examine group differences and to calculate effect sizes. As in the earlier trial of a “one shot” CBT-I treatment for individuals with acute insomnia [40], adherence was operationalized and calculated as an average number of minutes, from the first weeklong sleep diary following treatment, that the subject was outside their Prescribed Time in Bed and if this average was outside 15 min subjects were considered non-adherent. 

## 3. Results

There were 13 enquiries to the group treatment arm, by eligible subjects, whereas there were 15 to the individual treatment arm. However, one subject in the group treatment arm and two in the individual treatment arm did not complete the study. Of note, despite the posters advertising for individuals with acute insomnia there were a number of enquiries by individuals with chronic insomnia to both studies (*n* = 11 in the individual arm and *n* = 8 in the group arm). The final sample consisted of 19 females and six males. In the group treatment arm, there were 10 females and three males and in the individual treatment arm there were 9 females and 3 males. There were no differences between the groups in terms of age (group arm, Mean Age 39.62 ± 13.64; individual arm, Mean Age 42.00 ± 17.83) or sex (both at *p* > 0.05). There were no significant between group differences, at baseline, on any of the reported measures (all at *p* > 0.05) (see Table 1). 

A mixed ANOVA was used to examine group by time changes on scores on the primary outcome measure (i.e., Insomnia Severity Index Scores). There was no interaction effect (Wilks Lambda = 0.94, F(1,23) = 1.4, *p* > 0.05, partial eta squared = 0.06) or a main effect for group (F(1,23) = 0.03, *p* > 0.05, partial eta squared = 0.001) but there was a main effect for time (Wilks Lambda = 0.2, F(1,23) = 93.14, *p* < 0.001, partial eta squared = 0.8). Following a one-way multivariate analysis of variance on pre-post change scores for each of the variables of interest, psychometric scales (PHQ-9, GAD-7, ISI) and sleep diaries (TST, SL, WASO, SE) revealed the overall model was not significant (F(7,17) = 0.6, *p* > 0.05—Pillai’s Trace = 0.2, partial eta squared = 0.2). Further analysis of each scale and sleep diary item revealed no between-group differences on any dimension (all at *p* > 0.05).

In terms of a responder analysis, using the criteria of a score of ≤8 on the ISI, nine out of the 13 participants (69.23%) no longer met criteria for insomnia in the group treatment condition and nine out of 12 (75%) participants in the individual treatment group no longer met criteria for insomnia at the one-month follow-up. There were no significant between-group differences in effectiveness using this criterion (Fishers Exact X^2^ (1) = 0.1, *p* > 0.05). With regard to symptom reduction, in line with Morin and colleagues [36] (i.e., a reduction of 7 points on the ISI), nine out of 13 (69.23%) subjects in the group arm showed an improvement, whereas 10 out of 12 (83.33%) subjects in the individual arm showed an improvement. There was no difference in treatment response between groups (Fishers Exact X^2^ (1) = 0.68, *p* > 0.05).

Considering there were no apparent differences between the groups, Cohen’s d effect sizes were calculated, from pre-post means and standard deviations, for each variable of interest from the overall sample. The results show high effect sizes for PHQ-9 (*d* = 1.28), GAD-7 (*d* = 1.26), ISI (*d* = 2.27), Sleep Latency (*d* = 1.06), Wake after Sleep Onset (*d* = 1.01) and Sleep Efficiency (*d* = 1.23). As expected, the Cohen’s d for Total Sleep Time (*d* = 0.17) was small.

Finally, using the criteria of adherence (i.e., those, on average, who were within 15 min of their PTIB), 7 (53.85%) of the subjects in the group treatment arm were adherent versus 11 (91.67%) in the individual treatment arm. A chi-square demonstrated a significant difference between the groups (Fishers Exact X^2^ (1) = 4.43, *p* < 0.05) with those in the individual treatment arm demonstrating higher adherence levels than those in the group treatment arm.

## 4. Discussion

The present study sought to examine whether the “one shot” CBT-I could be used in a group treatment format and provide an, albeit preliminary, examination of the comparative efficacy and effectiveness of group therapy compared to individual therapy within the context of acute insomnia. 

In line with the hypothesis, the results suggest that in terms of the symptoms and treatment outcomes with respect to sleep and insomnia, with the exception of adherence, group treatment appears to be comparable to individual treatment. Whilst this finding is certainly not new within the context of chronic insomnia and the standard duration of CBT-I [27,47], this is interesting within the confines of acute insomnia and the “one-shot” treatment strategy. These findings suggest that it is plausible to deliver this intervention to individuals with acute insomnia in a very intensive manner (i.e., via groups) without reducing its efficacy or effectiveness. This naturally will aid in its potential for widespread dissemination and implementation in settings where it may be needed most (i.e., Primary Care), as it will be attractive to both patients and the agencies responsible for healthcare delivery. That said, there is concern with regard to levels of adherence in the group treatment arm, with just under half of the subjects meeting the criteria for being non-adherent. Interestingly, however, it would appear that levels of adherence have not impacted heavily on treatment outcome, although again sample size should be noted here. One possible explanation for this disparity in findings (non-adherence not influencing treatment outcome) is considering at present that there is no “gold standard” definition of adherence following CBT-I it may be that the criteria set down in the present study may have been too conservative and that variations outside this threshold of prescribed time in and out of bed do not impact on treatment outcome. Alternatively, it could be conceivable that the “sleep restriction” prescription, on which adherence rates were based in this study, may not be integral to treatment outcome in this population and a different measure of adherence is needed within the context of acute insomnia. Irrespective, adherence will need to be examined more closely in a larger trial, ideally a randomised control trial, of the intervention and if necessary the “one-shot” may need to be adapted to address adherence before widespread deployment.

Interestingly, the results suggest that, irrespective of delivery format (group or individual), the “one-shot” appears not only to confer a positive impact on insomnia severity and self-reported symptoms of insomnia (sleep latency and wake after sleep onset), but also impacts of the symptoms of anxiety and depression. Whilst this has been previously demonstrated in samples with chronic insomnia, using a “traditional” CBT-I approach [48,49], this is the first time these results have been shown within the context of a brief version of CBT-I or for those with acute insomnia. That said, these results should be viewed with a certain degree of caution as it is unknown whether this reduction in symptoms was due to the intervention or whether a level of natural remission occurs, in terms of the symptoms of anxiety and depression, over the early developmental course of insomnia. Sadly, the small sample size precluded a fine-grained analysis of this question.

The results should be viewed within the context of a few limitations. Primarily this was a relatively small self-selecting sample. More specifically, uptake was very low considering the suggested prevalence and incidence of acute insomnia in the general population [2]. Whether this speaks to the context of recruitment within community settings used in the present study or more broadly to a need to increase health literacy with regard to insomnia in general and acute insomnia specifically, so that appropriate pathways to prevention and treatment can be created, is unknown. Certainly it would be interesting to compare differences in uptake if the same study were to be advertised within the context of Primary Care as opposed to in community settings. Further, it should also be acknowledged that the groups were relatively small (i.e., four people) and so it is unknown whether the present findings would be translatable to bigger groups. That said, to date there is limited evidence on optimal group size within the context of the treatment of insomnia and as such it would be difficult to make any conclusions as to whether the small group size had a positive, or indeed negative, influence on treatment outcomes. Additionally, as subjects were not randomly allocated to either individual or group therapy we are unable to make any inferences about the acceptability of the different interventions. Certainly, the next step in this research agenda is to conduct a full RCT, which will include a no-treatment control group, additionally focusing on the acceptability of the different treatment modalities, for people with acute insomnia. Finally, the measure of adherence in the current study could be considered a limitation. Defining adherence on the basis of the first week of sleep diaries following treatment, although used previously [40], only provides data on initial adherence to the sleep restriction aspect of the overall CBT-I and nothing more. 

## 5. Conclusions

In summary, the present study demonstrates, albeit rather preliminarily, that a “one-shot” intervention for acute insomnia could be delivered in a group context without losing efficacy or effectiveness, at least compared to it being delivered individually. Moreover, the intervention, irrespective of its delivery modality, appears to confer benefits “above and beyond” sleep itself with an impact on the symptoms of depression and anxiety. However, issues of uptake and adherence will need to be addressed before wide scale dissemination and implementation can occur in Primary Care.

## Figures and Tables

**Figure 1 brainsci-07-00001-f001:**
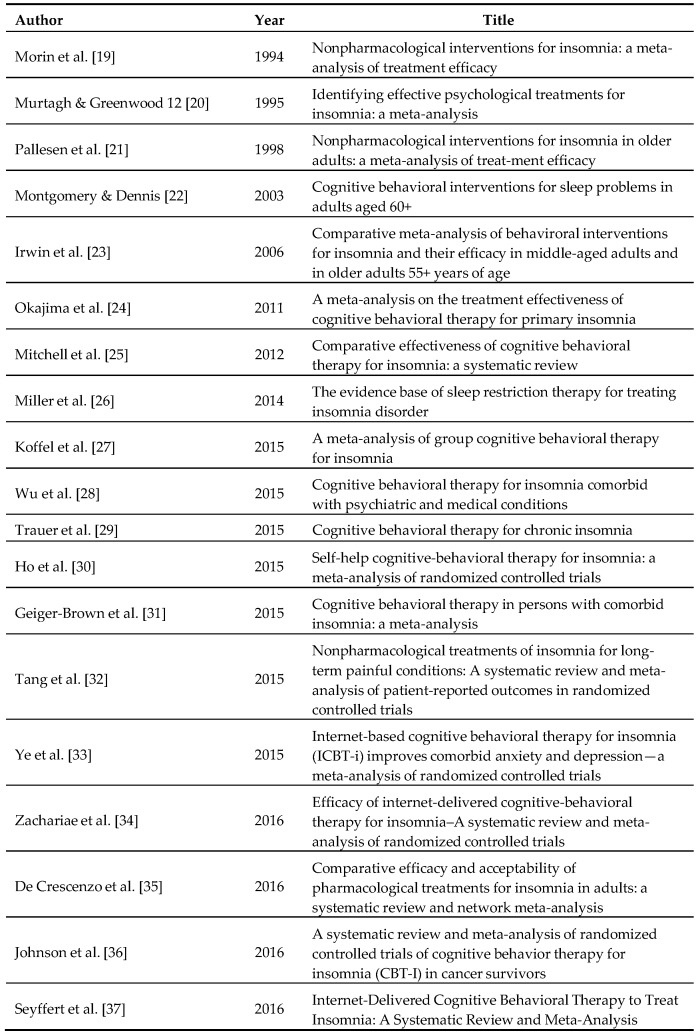
An overview of CBT-I meta-analyses.

**Table 1 brainsci-07-00001-t001:** Means and Standard Deviations by Group.

Variables	Baseline		Follow-Up		Change Scores
	Group (*n* = 13)	Individual (*n* = 12)	Group (*n* = 13)	Individual (*n* = 12)	
PHQ-9	7.39 (5.46)	6.92 (2.68)	2.15 (2.58)	3.5 (1.68)	−4.36 (3.92)
GAD-7	7.08 (5.14)	8.00 (3.44)	2.54 (2.6)	3.75 (1.91)	−4.4 (4.26)
ISI	15.92 (6.14)	14.5 (3.8)	5.15 (3.69)	6.08 (2.61)	−9.64 (5.01)
SL	27.95 (25.58)	28.53 (17.81)	10.8 (6.93)	12.24 (7.57)	−16.73 (19.81)
WASO	54.33 (35.63)	28.65 (26.98)	17.1 (13.92)	14.62 (14.89)	−26.11 (33.58)
SE	74.75 (13.54)	78.83 (9.81)	89.19 (4.83)	87.46 (7.58)	11.65 (13.21)
TST	397.55 (76.9)	425 (35.71)	407.47 (53.68)	433.45 (41.91)	9.21 (52.01)

PHQ-9: Patient Health Questionnaire; GAD-7: Generalized Anxiety Disorder; ISI: Insomnia Severity Index; SL: Sleep Latency; WASO: Wake After Sleep Onset; SE: Sleep Efficiency; TST:Total Sleep Time.

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
