# Peer review of "Group vs. Individual Treatment for Acute Insomnia: A Pilot Study Evaluating a “One-Shot” Treatment Strategy"

_brainsci, 2016, doi:10.3390/brainsci7010001_

Round 1

Reviewer 1 Report

The introduction is clearly written with a clear rationale. The methods section is also clearly laid out. However, the results and discussion need some attention.

One potential problem with the design is that it is not a randomised controlled trial. The study is on two different groups recruited with somewhat different understandings of the nature of the study, rather than one recruitment poster and then a randomisation to the two conditions. Therefore, one cannot be sure that the degree of self selection, motivation, or understanding of the nature of the study is comparable between the two groups. Any differences in these effects may affect the degree of adherence or outcome benefits of the therapies. (Perhaps an overzealous ethics committee may have dictated the use of different recruitment posters without appreciating the potential limitation of directly comparing between the two groups). In any case this needs to be raised as a possible limitation of the design in the discussion section.

Other specific comments will refer to line numbers in the ms.

line 135; some error here, should it read, "told to stick with .  .  "

lines 193,4; I believe it is more conventional to refer to non significant differences with p>0.05.

lines 197,8; The use of "one-way" MANOVAS and ANOVAS is puzzling. The "model" should be an exploration of the significance of the interaction term in a 2-way ANOVA with main effects of group and time (baseline vs follow-up) to test diffences between groups in amount of change between times.

line 199; What is meant by the "overall model was not significant"? What was the model? testing the interaction term?

lines 202-5; This is conventionally called a "responder analysis"

lines 206-9; What test was used for comparing between percentages responding?

line 221-32; I don't think this paragraph contributes anything worth reporting. The "uptake" is similar but you have no data on how many people read each poster nor how many potential acute insomniacs read each poster. If there were, for example, 50 eligible participants who read the group poster and only 20 who read the individual treatment poster you could conclude a differential take up. But without that data, this paragraphs speculation is empty.

line 235; possible typo? should read " is certainly not new within . . ."?

lines 237-8; The suggestion is that this CBT-I in "one shot" does not lose effectiveness. However, to conclude that would require a direct comparison with CBT-I modes with more than one session. At the very least one could compare effect sizes with those of multiple session treatments from meta-analyses.

line 241; "commissioners of healthcare services." Perhaps a more generic description can be used than commissioners, "agencies responsible for health care delivery"?

line264; I would say that sample size is not the problem here, but the lack of direct comparison with a non-treatment control and CBT-I with more sessions.

Author Response

Dear Reviewer,

Thank you for your comments on the manuscript. We believe have addressed each comment and feel that with your suggestions the manuscript is now much more robust. We have highlighted our responses in green:

The introduction is clearly written with a clear rationale. The methods section is also clearly laid out. However, the results and discussion need some attention.

Response – We thank the reviewer for their comments and hope the results and discussion are now more appropriate after taking on board the reviewers comments.

One potential problem with the design is that it is not a randomised controlled trial. The study is on two different groups recruited with somewhat different understandings of the nature of the study, rather than one recruitment poster and then a randomisation to the two conditions. Therefore, one cannot be sure that the degree of self selection, motivation, or understanding of the nature of the study is comparable between the two groups. Any differences in these effects may affect the degree of adherence or outcome benefits of the therapies. (Perhaps an overzealous ethics committee may have dictated the use of different recruitment posters without appreciating the potential limitation of directly comparing between the two groups). In any case this needs to be raised as a possible limitation of the design in the discussion section.

Response – Yes, we agree with the reviewer and have explicitly stated in the discussion that we cannot make any inferences about the acceptability of the interventions, just their comparative efficacy and effectiveness. We have also removed any reference to the text with regard to the uptake of the interventions (Lines 238-246). Our intention for the RCT will be to randomly allocate participants to either group, individual or wait-list control and we have outlined that the study was not an RCT as a limitation in the discussion (Lines 292-299).

Other specific comments will refer to line numbers in the ms.

line 135; some error here, should it read, "told to stick with .  .  "

Response – We apologise. Yes this was a typo and we have changed the text accordingly (Line 137).

lines 193,4; I believe it is more conventional to refer to non significant differences with p>0.05.

Response – We have revised the manuscript and referred to non-significant results in the text as suggested.

lines 197,8; The use of "one-way" MANOVAS and ANOVAS is puzzling. The "model" should be an exploration of the significance of the interaction term in a 2-way ANOVA with main effects of group and time (baseline vs follow-up) to test diffences between groups in amount of change between times.

Response – We have added a Mixed ANOVA on the primary outcome measure (ISI) to examine the time x group interaction (Lines203-207). We hope this is okay?

line 199; What is meant by the "overall model was not significant"? What was the model? testing the interaction term?

Response – The overall model was examining group differences in change scores on each of the variables of interest (psychometrics and sleep diary data) and as such the overall model not being significant suggests that combined these variables failed to differentiate between the groups. We have made this more explicit in the text (Lines 184-187)

lines 202-5; This is conventionally called a "responder analysis"

Response – Thank you, we have now included the term in the text (Line 212).

lines 206-9; What test was used for comparing between percentages responding?

Response – Thank you for highlighting this. We used a Chi-Square to compare treatment responders versus non responders and have now included the results in the manuscript (Lines 219).

line 221-32; I don't think this paragraph contributes anything worth reporting. The "uptake" is similar but you have no data on how many people read each poster nor how many potential acute insomniacs read each poster. If there were, for example, 50 eligible participants who read the group poster and only 20 who read the individual treatment poster you could conclude a differential take up. But without that data, this paragraphs speculation is empty.

Response – We agree and have removed this paragraph (Lines 238-246).

line 235; possible typo? should read " is certainly not new within . . ."?

Response – We apologise for the typo and have changed the text accordingly (Line 249).

lines 237-8; The suggestion is that this CBT-I in "one shot" does not lose effectiveness. However, to conclude that would require a direct comparison with CBT-I modes with more than one session. At the very least one could compare effect sizes with those of multiple session treatments from meta-analyses.

Response – We apologise for not being clear. We suggest here that using the one shot in the context of group therapy does not reduce its effectiveness compared to when it is used in individualized therapy for people with acute insomnia. We have now made this more explicit in the text (Lines 251-253).

line 241; "commissioners of healthcare services." Perhaps a more generic description can be used than commissioners, "agencies responsible for health care delivery"?

Response – We agree with the reviewer’s suggestion and have changed the text accordingly (Lines 255-256).

line264; I would say that sample size is not the problem here, but the lack of direct comparison with a non-treatment control and CBT-I with more sessions.

Response – We agree and have added a section into the limitations section to exemplify this point (Lines 292-299).

We look forward to hearing your thoughts on the revised manuscript.

Best wishes,

Jason Ellis, Pam Boullin and Christina Ellwood

Reviewer 2 Report

I was happy to read this well done and well described pilot study of early intervention to prevent development of chronic insomnia. Prevention of chronic insomnia is important because of the large population, and as a possible means to prevent development of depression and other insomnia-related disorders. I believe this type of brief and effective intervention, going beyond the usual pamphlet about sleep hygiene, is much sought after and needed in primary care. I very much look forward to a larger study, preferably set in primary care or similarly available for that patient group. I only have a few minor remarks and questions.

Abstract: Please add the results in the form of the effect size of ISI.

Row 67: Reference 20 appears to be referring to the wrong paper, I guess it should be: Ellis, Jason G., et al. "The natural history of insomnia: acute insomnia and first-onset depression." Sleep 37.1 (2014): 97.

Row 182: The adherence criteria is interesting. Is there a precedence for this measure? Please explain the adherence criteria in more detail. Was this the diary of the 4th week? How was the Prescribed TIB decided, and by whom? Do you have daily sleep diary data for all/most participants from all four weeks so that you can check adherence over the course of the treatment (perhaps those with poor adherence week 4 adhered perfectly earlier in the treatment)? Would you, for the next study, choose to do it differently in any way or do you/the therapists think it was a good representation of adherence on the whole?

Row 189: Please include number of interested individuals with chronic insomnia, this is of interest for the overall mission of CBT-i dissemination research.

Row 202: Recommended criteria for remission measured with ISI after treatment is < 8 p (Morin & Espie 2003) as opposed to detecting insomnia where > 10 p is recommended. You may want to use < 8 p only for the purposes of the paper, and/or report response rates (>7 p reduction).

Row 226: Couldn’t one explanation be that an individual only saw one of the posters and applied to the arm represented there, rather than choosing this mode of delivery over the other? If I understand correctly the posters were not displayed side by side? For the next study I propose asking the participants of their preference, before and after treatment.

Row 235: I think you mean “is certainly not new”?

Discussion: A suggestion for an upcoming RCT is to investigate what the participants learn about CBT-i methods, and the relation to (long-term) effects. That could perhaps reveal more about what type of adherence is important.

Author Response

Dear Reviewer,

Thank you for your comments on the manuscript. We believe have addressed each comment and feel that with your suggestions the manuscript is now much more robust. We have highlighted our responses in green:

I was happy to read this well done and well described pilot study of early intervention to prevent development of chronic insomnia. Prevention of chronic insomnia is important because of the large population, and as a possible means to prevent development of depression and other insomnia-related disorders. I believe this type of brief and effective intervention, going beyond the usual pamphlet about sleep hygiene, is much sought after and needed in primary care. I very much look forward to a larger study, preferably set in primary care or similarly available for that patient group. I only have a few minor remarks and questions.

Response – We thank the reviewer for their kind comments regarding the present study and their enthusiasm for our programme of work

Abstract: Please add the results in the form of the effect size of ISI.

Response – We have now included the effect size of the ISI in the abstract (Lines 26-28)

Row 67: Reference 20 appears to be referring to the wrong paper, I guess it should be: Ellis, Jason G., et al. "The natural history of insomnia: acute insomnia and first-onset depression." Sleep 37.1 (2014): 97.

Response – We apologise. Yes, that was not the correct reference and we have changed this accordingly (reference 20 and reference 2 have been swapped)

Row 182: The adherence criteria is interesting. Is there a precedence for this measure? Please explain the adherence criteria in more detail. Was this the diary of the 4th week? How was the Prescribed TIB decided, and by whom? Do you have daily sleep diary data for all/most participants from all four weeks so that you can check adherence over the course of the treatment (perhaps those with poor adherence week 4 adhered perfectly earlier in the treatment)? Would you, for the next study, choose to do it differently in any way or do you/the therapists think it was a good representation of adherence on the whole?

Response – This is an interesting point. To our knowledge there is no accepted measure of adherence following CBT-I and the most widely used method, at present, is to use sleep diary data to infer about adherence. As such, we have used the same criteria that was used in Ellis et al’s (2015) RCT to maintain parity between studies and have made this more explicit in the method section. We have also highlighted this as a potential limitation in the discussion (Lines 186-189).

Row 189: Please include number of interested individuals with chronic insomnia, this is of interest for the overall mission of CBT-i dissemination research.

Response – We apologise and have now included the rates of enquiry by people with chronic insomnia. That said, as the poster specifically asked for individuals with acute insomnia, sadly, I am not sure the extent to which that data can inform overall CBT-I dissemination research (Lines 193-196).

Row 202: Recommended criteria for remission measured with ISI after treatment is < 8 p (Morin & Espie 2003) as opposed to detecting insomnia where > 10 p is recommended. You may want to use < 8 p only for the purposes of the paper, and/or report response rates (>7 p reduction).

Response – We agree with the reviewer regarding the cut-off criteria for the ISI and have removed the section using the >10 criterion. We also agree that including the improvement rates would be beneficial and have done so based upon Yang et al’s 2009 criteria of a reduction of >7 points on the ISI for a minimally important difference (Lines 212-223).

Row 226: Couldn’t one explanation be that an individual only saw one of the posters and applied to the arm represented there, rather than choosing this mode of delivery over the other? If I understand correctly the posters were not displayed side by side? For the next study I propose asking the participants of their preference, before and after treatment.

Response – Yes, we agree with the reviewer and have explicitly stated in the discussion that we cannot make any inferences about the acceptability of the interventions, just their comparative efficacy and effectiveness (Lines 292-294). Moreover, we have removed any reference to examining uptake (Lines 238-246). Our intention for the RCT will be to randomly allocate participants to either group, individual or wait-list control.

Row 235: I think you mean “is certainly not new”?

Response – We apologise for the typo and have changed the text accordingy (Line 249).

Discussion: A suggestion for an upcoming RCT is to investigate what the participants learn about CBT-i methods, and the relation to (long-term) effects. That could perhaps reveal more about what type of adherence is important.

Response – Thank you. That is an excellent suggestion and as we are planning to do a full RCT next it would really add to determine, presumably through interviews, what participants learn and can recollect regarding their experiences of CBT-I and see how that impacts on long-term outcomes.

Best wishes,

Jason Ellis, Pam Boullin and Christina Ellwood

Reviewer 3 Report

This paper reports on a small clinical trial (N = 28) comparing the efficacy of a single CBT-I session provided individually or in a group format for acute insomnia. The research questions are of interest, the methodology is reasonable, and the findings are generally positive.  There are a few issues, however, that might warrant further consideration to improve the impact of the paper.  First, the lack of a “no treatment” control group restricts significantly the scope and interpretation of the findings. Although it is of interest to show that group therapy is not inferior to individual therapy, the real question of interest for a study like this one on acute insomnia would be whether an early intervention reduces the risk of chronic insomnia and/or the development of clinically meaningful depressive symptoms.  The ideal study design to test this question would have been the comparison of a group receiving treatment during the acute phase of insomnia against another group that did not; obviously, this would have required a much larger sample size than what was used in this pilot study. In any case, I think this issue warrants further discussion. Secondly, I disagree with the conclusion that the present study suggests that group therapy is as acceptable as individual therapy.  Just counting the number of participants recruited with each advertisement location does not provide an adequate test of treatment acceptability because of several potential confounding factors (i.e., visibility, SES neighborhood, etc.) that may have impacted on response rates.  A better strategy would have been to use the same ad in all places but then ask potential participants to provide preference ratings for the two treatment modality. Thirdly, the usual reporting period for the ISI is either “last two weeks” or “last month”; it can be used for a “last week” time period but has not been validated for this type of reference period.  A final point is with regard to the 20%-30% drop out rates mentioned in the introduction.  This number does not fit at all with my reading of the CBT-I literature which would suggest a drop out rate closer to 10% than 30%; please double check these figures. Overall, this is a very good paper that could be improved by addressing these issues.

Author Response

Dear Reviewer,

Thank you for your comments on the manuscript. We believe have addressed each comment and feel that with your suggestions the manuscript is now much more robust. We have highlighted our responses in green:

This paper reports on a small clinical trial (N = 28) comparing the efficacy of a single CBT-I session provided individually or in a group format for acute insomnia. The research questions are of interest, the methodology is reasonable, and the findings are generally positive.  There are a few issues, however, that might warrant further consideration to improve the impact of the paper.  First, the lack of a “no treatment” control group restricts significantly the scope and interpretation of the findings. Although it is of interest to show that group therapy is not inferior to individual therapy, the real question of interest for a study like this one on acute insomnia would be whether an early intervention reduces the risk of chronic insomnia and/or the development of clinically meaningful depressive symptoms.  The ideal study design to test this question would have been the comparison of a group receiving treatment during the acute phase of insomnia against another group that did not; obviously, this would have required a much larger sample size than what was used in this pilot study. In any case, I think this issue warrants further discussion.

Response – We agree that an ideal design would have been a RCT with a wait-list control and we have added a section regarding this into the limitations section (lines 294-299). Certainly the next stage in this research agenda is, now we have demonstrated that it can be used in a group context, to conduct a full RCT.

Secondly, I disagree with the conclusion that the present study suggests that group therapy is as acceptable as individual therapy.  Just counting the number of participants recruited with each advertisement location does not provide an adequate test of treatment acceptability because of several potential confounding factors (i.e., visibility, SES neighborhood, etc.) that may have impacted on response rates.  A better strategy would have been to use the same ad in all places but then ask potential participants to provide preference ratings for the two treatment modality.

Response – We agree with the reviewer that the word ‘acceptable’ is probably not appropriate considering the context of the study. As such we have explicitly stated in the discussion that we cannot make any inferences about the acceptability of the interventions, just their comparative efficacy and effectiveness (Lines 292-294). Additionally, we have deleted any reference to examining uptake in the manuscript (Lines 238-246).

Thirdly, the usual reporting period for the ISI is either “last two weeks” or “last month”; it can be used for a “last week” time period but has not been validated for this type of reference period. 

Response – We agree with the reviewer that the ISI has not yet been validated for use on a one-week assessment. That said, whilst it is the case that the ISI was designed on the basis of a retrospective assessment of the previous month, due to the timeframe of the intervention we felt that a one-month retrospective assessment would not capture well whether the intervention had a positive outcome on the subjects’ symptoms. Additionally, we wished to retain parity with the existing literature on the treatment of acute insomnia using the ‘one shot’ with the timeframe of one week. We are in the process of validating the ISI within the context of a week.

A final point is with regard to the 20%-30% drop out rates mentioned in the introduction.  This number does not fit at all with my reading of the CBT-I literature which would suggest a drop out rate closer to 10% than 30%; please double check these figures. Overall, this is a very good paper that could be improved by addressing these issues.

Response – This is a very interesting point. Whilst the seminal study by Ong, Kuo and Manber, which we reference, does suggest smaller drop out rates (0-8%) within the context of two previous RCTs they also point out drop out rates ranging from (9.7-38.3%) in clinical settings. Further, in their own analysis 39.96% of the subjects dropped out. As such we chose a mid-range of 20-30%. We have now removed the numbers and added a caveat that these high levels are especially seen in clinical settings (Lines 52-53).

If there is any further information you require, please do not hesitate to contact us.

Best wishes,

Jason Ellis, Pam Boullin and Christina Ellwood

Round 2

Reviewer 1 Report

The three reviewers showed some consistency in their comments about relatively minor points in the manuscript and the authors have done a very good job of responding to those comments and making amendments.

The article is in a publishable form in my opinion.